# "What's Past Is Prologue": Vegetation Model Calibration with and without Future Climate

**Ellynne Kutschera** [1,*]📷, **John B. Kim** [2]📷, **G. Stephen Pitts** [3] **and Ray Drapek** [2]📷

1 USDA Forest Service Pacific Northwest Research Station ORISE Fellow, 3200 SW Jefferson Way, Corvallis, OR 97331, USA

2 Forest Service Western Wildland Environmental Threat Assessment Center, 3200 SW Jefferson Way, Corvallis, OR 97331, USA; john.kim@usda.gov (J.B.K.)

3 College of Forestry, Oregon State University, 3100 SW Jefferson Way, Corvallis, OR 97331, USA; bear.pitts@oregonstate.edu

* Correspondence: ellen.kutschera@usda.gov

**Abstract:** Many models are designed to generate future predictions under climate-change scenarios. Such models are typically calibrated for a study area with climate data that represent historical conditions. However, future projections of the model may include outputs for which the model has not been calibrated. Ideally, a climate-change-impacts model would be calibrated for recent conditions and also for possible future climate conditions. We demonstrate an approach, where a vegetation model is subjected to two calibrations: conventionally to the study area and separately to the study area plus additional areas representing analogues of potential future climate. We apply the dynamic vegetation model MC2 to a mountainous ecosystem in the Pacific Northwest, USA. We compare the conventional model calibration with the extra-study-area calibration and future projections. The two calibrations produce different outputs in key ecosystem variables, where some differences vary with time. Some model output trends for net primary productivity and plant functional type are more influenced by climatic input, while for others, the calibration area has greater consequence. Excluding areas representing potential future climate may be an important omission in model calibration, making the inclusion of such areas a decisive consideration in climate-change-impact simulations.

**Keywords:** model calibration; DGVM; MC2; climate change; climate analogues; earth-systems modeling

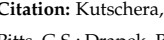



## 1. Introduction

As the pursuit of future predictions for earth systems continues, many models simulate present and future impacts of changing climate. Whether modeling hydrological systems, fire, biotic populations, or other aspects of earth systems, calibration and study-area selection are integral to model analysis [1–4]. Often, earth-systems models are calibrated to a particular study area [5–8]. For example, Ref. [8] simulated vegetation in the Blue Mountains ecoregion in Oregon, USA, and calibrated the vegetation model to the specific study region of 7.2 Mha. Ref. [7] applied a land surface model for high latitudes calibrated with region-specific scaling factors and vegetative parameters. However, for process-based models, calibration of this sort may be characterized by non-stationarity, or the problem of implicitly assuming that estimated average statistical values for parameter relationships will remain unchanged when the model is projected into a future time or at a different place [9]. Rather than constraining the model to the range of data on which its estimates are based, the model is applied to domains for which the parameter relationships may no longer hold. In general, projecting into conditions to which a model has not been calibrated may generate unreliable results.

The Coupled Model Intercomparison Project Phase 6 (CMIP6) outlines an array of forcing scenarios for the future, known as shared socioeconomic pathways (SSPs) [10]. Included

in these are several future pathways of higher emissions and substantial land-use change that represent potential dramatic changes in climate conditions. SSP5-8.5 is a scenario where economies are energy intensive and fossil based, potentially resulting in a forcing level of 8.5 W m$^{-2}$ by the year 2100. Additionally, carbon-cycle feedback may be most prominent under this scenario. When projecting changes into the future, such future climate conditions driving an earth-systems model may not be present in the climate data for calibration and evaluation of model performance. Even for pathways that are less fossil based, combinations of factors, such as forcing due to aerosols and land-use-change–atmospheric chemistry interactions can cause strong impacts on climate. Net-positive carbon-cycle feedback can push radiative forcing and lead to climatic disruptions. If climatic impacts are not adequately modeled, drought, fire, vegetation changes, and other consequences for ecosystems can be wrongly understood or missed. With drought increases, for example, knowledge of how ecosystems respond and the biophysical processes underlying drought are sought by modelers and planners to address changes in biodiversity [11]. Earth-system models may be simulating responses to climate conditions that are novel compared to calibration conditions. When drawing conclusions from the model output, the analysis may rely on unknown model performance.

To our knowledge, a thorough discussion of this aspect of model calibration is not present in the scientific literature. Many researchers have addressed calibration methods to improve model uncertainty, although this tends to be within the bounds of the known parameters of the model or perturbations in the vicinity thereof [12,13]. The transferability of models, or a model's ability to be transferred to novel conditions, has been discussed specifically for ecological models [14]. While a range of tools has been suggested for quantifying novelty for varying distributions to which ecological models may be applied, testing the predictive performance of models transferred into novel conditions is uncommon. We surmise that the difficulty lies in model comparison to unknown conditions.

Model calibration is particularly important to dynamic global vegetation models (DGVMs), models that simulate the terrestrial biome by computing both ecosystem processes and the transition of vegetation types through time. While DGVMs can be run over the extent of the globe, especially for ensemble or comparative studies [15], they have also been applied to many regions and subregions for regional and localized modeling. Ref. [16] ran the DGVM MC2 for the Upper Heihe River Basin, a mountainous plateau region with substantial climate variation. Ref. [17] evaluated the performance of 14 DGVMs for the Southwestern US across a range of ecosystem types. DGVMs may be tuned to these specific regions, increasing the sensitivity of the model to the regions at the exclusion of exterior conditions. While climate-change scenarios themselves are projections and not predictions, improving model-calibration methods for a larger range of conditions could improve the plausibility of future projections for climate-change impacts.

In this paper, we present an approach to address the deficiency of model calibration with respect to climate analogues. In our method, we calibrate our model not only to the study area but to areas that represent possible future conditions of the study area. We run a DGVM for a specific ecoregion as the study area to explore the differences in model performance under two conditions. In the first, the model is calibrated only to the study area. In the second, the model is calibrated to a larger domain that includes both the study area and areas currently experiencing climate conditions that may appear in the study area in the future. We present both a methodology and the results from its application.

To assess the inclusion of extra-study-area-domain-representing climate analogues on model calibration and future projections, we consider the following questions:

1. What differences appear in the final calibrations of the extra-study-area calibration compared to the conventional calibration?
2. What differences in future output occur in the extra-study-area model compared to the conventionally calibrated model?
3. What factors drive the differences?

By comparing the conventional calibration approach with the new, we demonstrate that the inclusion of climate-change-analogue areas outside the focus study area has important consequences for model calibration and future model projections.

## 2. Materials and Methods

### 2.1. Study Area

The study area is the Blue Mountains Ecoregion (BME), which ranges over 7.2 Mha of mountains and basins in Eastern Oregon, USA (Figure 1). A Level III Ecoregion Environmental Protection Agency designation defines the area [18]. Climatologically diverse, common in BME are dry conifer forests interspersed by shrublands, dry grasslands, and shrub steppe, where dense forests of mixed conifers dominate at higher elevations [19,20]. Temperatures range from −5 to 5 °C in the winter and up to 5.5 to 18 °C in the summer. Precipitation varies from 20 to 100 cm annually. Lower elevations are found toward the southern and western parts of BME. Common species at lower elevations include ponderosa pine (*Pinus ponderosa* Douglas ex P. Lawson and C. Lawson), western juniper (*Juniperus occidentalis* Hook.), Idaho fescue (*Festuca idahoensis* Elmer), bitterbrush (*Purshia tridentata* [Pursh] DC.), curl-leaf mountain mahogany (*Cercocarpus ledifolius* Nutt.), and various sagebrushes. Higher-elevation, mountainous areas occur in the central and northern sections of BME. The Wallowa mountains are found to the northeast and range from 900 up to 3000 m. At only slightly lower elevation is the Umatilla National Forest that extends to the west and north, reaching into Washington, with Malheur National Forest to the south. Soils are volcanic in origin, with wetter ash soils at high elevations supporting dense forests [19]. Common species at higher elevations include lodgepole pine (*Pinus contorta* var. *latifolia* Engelm. ex S. Watson), Douglas fir (*Pseudotsuga menzieii* (Mirb.) Franco), grand fir (*Abies grandis* (Douglas ex D. Don) Lindl.), and western larch (*Larix occidentalis* Nutt.). Species seen at the highest elevations include subalpine fir (*A. lasiocarpa* (Hook.) Nutt.) and whitebark pine (*P. albicaulis* Engelm.), where alpine meadows of sedge and fescue also occur.

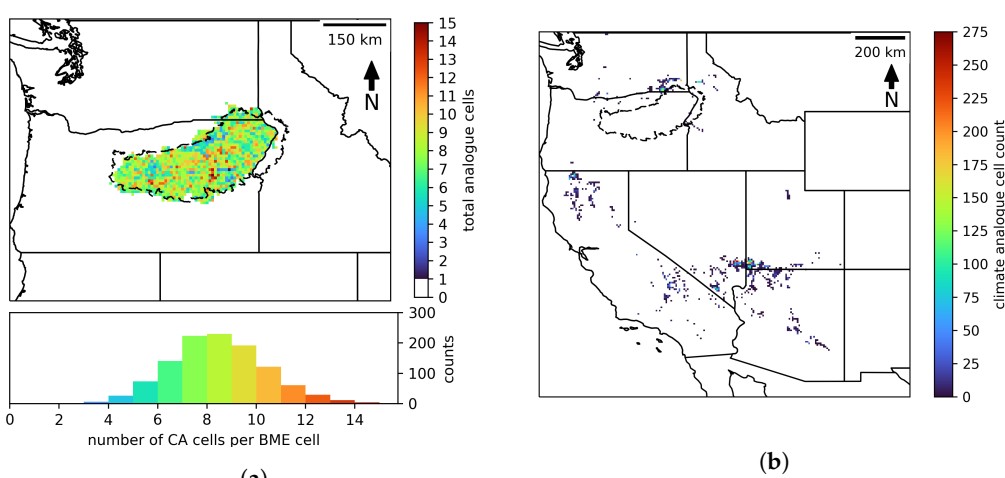

**Figure 1.** Study area is in (**a**) showing the number of climate-analogue (CA) cells from CSIRO-Mk3.6.0 for each Blue Mountains Ecoregion (BME) cell. Calibration domain in (**b**) is the set of CA cells for BME determined from climate input for the global climate model (GCM) CSIRO-Mk3.6.0. The study area is mapped showing only the calibration cells, sampled every 10th cell along latitude and longitude from the full 30 arc-second simulation grid. The histogram shows the number of CA cells associated with each sampled cell within the BME study area.

### 2.2. Extended Model Calibration Domain

The model for our study is the DGVM MC2. Conventionally, MC2 represents the simulation domain as a grid at a 30 arc-second resolution (approximately 800 m), comprising 115,914 grid cells for BME. For the climate-analogue selection process, the grid was limited to every 5th grid cell for computational tractability. In total, 28 global climate models

(GCMs) provided climate input data. For a given GCM, a future MC2 simulation was run to the end-of-century; from the output, 30 grid cells within the Western USA per one pixel of BME were identified that are climate analogues of the future (2070–2099) climate of the BME grid cell. In theory, the present day climate of the analogue cells represents the future climate of the grid cell within the BME study area. Climate-analogue (CA) cells were determined according to the standardized Euclidean distance ($D_{SED}$) method described by [21]. We calculated the dissimilarity metric $D_{SED}$ for BME CA cells on the basis of average precipitation and average temperature from the historical period 1970–1999 to the future period 2070–2099. PRISM climate data [22] provided the historical values, and future values were taken from GCM climate data. The 30 closest CA cells for each BME cell from the CA $D_{SED}$ calculation were selected. CA cells are not mutually exclusive, meaning that a cell that is a CA cell for one cell in BME may be a CA cell for another BME cell.

Computational times for calibration that included all 30 CA cells for each BME cell from each GCM were prohibitively long. As an alternative, we created a subset of CA cells for calibration by sampling every other cell along both latitude and longitude, reducing the grid to one-quarter of the resolution. Consequently, in our final calibration, each BME study area cell has 1–16 CA cells with an average of 7.5 CA cells across all CA cells from all GCMs. The total CA cells per GCM under the reduced grid is approximately 8,600 (some geographically overlapping), which is all CA cells for BME for the given GCM. Figure 1a shows the set of CA cells for all BME cells for climate input from the GCM CSIRO-Mk3.6.0. As shown, some CA cells are CA cells for more than 200 BME cells. Figure 1b sums the total number of CA cells for each BME cell and gives the distribution of the number of CA cells per BME cell. For all CA cells from all GCMs averaged together, each BME cell is associated with an average of 7.5 CA cells that may be within or outside the BME boundary. Maps are at the reduced grid resolution.

For our calibration study, we examined two areas: the first being only the BME region and the second including the BME region and the CA cells. Henceforth, we refer to the two calibrations and their results as "BME-only" and "BME-plus", respectively. Each set of cells represents a different range of temperature and precipitation values associated with the respective calibration area. As seen in Figure 2, the temperature and precipitation input range for BME-plus extends significantly beyond that of BME-only.

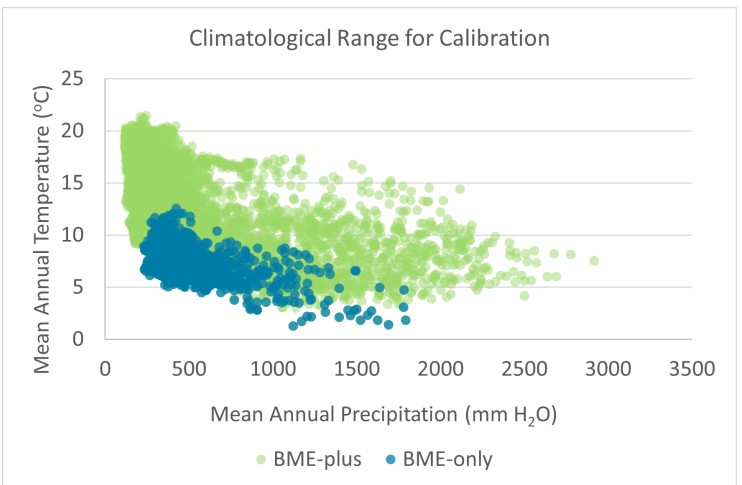

**Figure 2.** Mean annual temperature vs. mean annual precipitation from PRISM climate data [22], 1983–2012, for the set of calibration cells in BME-plus and BME-only (reduced grid).

### 2.3. MC2 Dynamic Global Vegetation Model

The DGVM MC2 is documented in detail elsewhere [23,24], and the code is available online [25]. Here, we describe salient features of the model. MC2 represents the land surface as a grid and runs at a monthly time step. Specifically, carbon and nutrient dynamics are simulated at a monthly resolution. At its finest scale, MC2 represents the simulation

domain as a grid at a 30 arc-second resolution (approximately 800 m). The model employs a modified version of the CENTURY soil organic model ([26]), the MC-Fire model for fire simulation ([24]), and a biogeography module based on the MAPSS model ([27]). Input to MC2 are climate data as monthly averages, elevation, soils data, and soil bulk density. Soils data for our study were taken from the State Soil Geographic Database (STATSGO) and Soil Survey Geographic Database (SSURGO) [28,29]. The CENTURY module simulates biogeographical processes by simulating carbon and water flow through the ecosystem. It maintains multiple carbon pools, including tree, grass, litter, and soil organic matter. Trees and grass compete for soil, water and light. Vegetation growth is calculated directly as the net primary production (NPP) as a function of temperature and available soil moisture, without calculating the gross primary productivity and autotrophic respiration. The fire module calculates fuel loading, fuel moisture, fire occurrence, and fire behavior, as well as the effects of fire occurrence. The biogeography module applies rules annually that classify each grid cell into 1 of 5 major biome types and 1 of 25 plant functional types (PFTs). The PFT of each grid cell subsequently affects fire simulations.

Four distinct phases comprise MC2 simulation. In the spin-up phase, the model's internal states are set using a 30-year monthly climatology representing early 1900s climate. In the second phase, the model is brought to dynamic but steady-state conditions with transient climate representing inter-annual variability from 1895 to 2008 without the long-term trend. Third, a historical simulation is run from 1895 to 2008 based on PRISM climate data [22]. Finally, future simulations are run from 1950 to 2100 with NEX-DCP30 GCM data [30] for the representative concentration pathway (RCP) 8.5 climate change scenario [31] at a 30 arc-second spatial resolution. We ran one future simulation for each of the 28 GCMs included in the NEX-DCP30 set. Fire suppression was applied for both historical and future runs.

MC2 code is a hybrid of C++ and Fortran (the CENTURY module). Simulations were run on a high-performance computing platform with six parallel processing nodes. Each node is a Dell R440 Poweredge Server with Intel Gold 5118 2.3 GHz processors that comprise 24 cores with 2 threads per core. Simulations typically take approximately 8 h to run from spinning up initial conditions to the present day at full spatial resolution. Future simulations run approximately 2.5 h per GCM. For a single calibration run at the reduced grid, the simulation time is approximately 6–8 min. Given the availability of computing resources, calibration at the reduced grid made executing over a hundred or more calibration runs feasible.

### 2.4. Model Calibration

Running MC2 for the study area at a 30 arc-second resolution resulted in long computing times, where each simulation of the historical period took hours on a high-performance computing platform. To run the model quickly and repeatedly for calibration, we sampled the grid to every 10th grid cell along the latitude and longitude for calibration, culminating in a calibration of 1120 cells in the BME study area and 6860 CA cells outside the BME study area (some CA cells appear within the BME region itself). Thus, the calibration resolution was at 300 arc-seconds or 5 arc-minutes. A comparison of fine resolution (30 arc-second) simulations and coarse (5 arc-minute) simulations have virtually identical regional summaries [5].

We calibrated MC2 twice, following [8]. In one calibration, we calibrated MC2 for the study domain only (BME-only); in the other, we calibrated MC2 for the study area plus the larger domain of the CA areas (BME-plus). We adjusted multiple variables (Table 1) so that the simulation output for the historical period had acceptable agreement with the benchmark data. For NPP, we referenced the MODIS Terrestrial Gross and Net Primary Production Global Data Set, version MOD17 [32]. Two potential natural vegetation maps ([33,34]) provided benchmark data for biome and PFT. For aboveground carbon stocks, we referenced the NACP Aboveground Biomass and Carbon Baseline Data (NBCD) [35]. We

calculated the mean fire return interval (MFRI) from the area burned as simulated by MC2 and compared the result with the LANDFIRE MFRI dataset [36].

**Table 1.** Calibrated parameters for MC2 when the model is calibrated to the study area only (BME-only) and to an extended calibration domain (BME-plus). Parameter values are for calibration with both fire and fire suppression enabled.

| Parameter | BME-only | BME-plus |
|---|---|---|
| maximum Net Primary Production (g C m$^{-2}$ mo$^{-1}$) | | |
| EN (evergreen needleleaf) | 250 | 300 |
| DB (deciduous broadleaf) | 180 | 180 |
| EB (evergreen broadleaf) | 180 | 180 |
| DN (deciduous needleleaf) | 250 | 250 |
| carbon thresholds (g C m$^{-2}$) | | |
| desert grass max C | 330 | 385 |
| desert tree max C | 330 | 80 |
| grass fraction threshold | 0.75 | 0.78 |
| woodland carbon threshold | 1100 | 900 |
| forest carbon threshold | 1450 | 1300 |

*2.5. Model Evaluation*

While there are no future observations that may be applied to evaluate future model projections, we interpreted CA cells as an approximation of future conditions. We developed a process to generate a "benchmark" of PFT for future output comparison. At the reduced calibration grid, for each GCM individually, we mapped the CA cells to the Küchler benchmark map for biome. We note that the Küchler benchmark map is for potential natural vegetation; MC2 simulates the potential vegetation by design, although this representation of vegetation without human intervention may be a limitation of the climate-analogue process. We then selected each group of CA cells associated with one BME cell for all BME cells and found the biome mode for each group. The result was a biome classification for each cell of BME taken from the biome mode of the CA cells associated with each BME cell. We thus generated a "future" biome map of BME for each GCM. Understandably, the future "benchmark" is dependent on climate-analogue theory and derivations and should not be considered a prediction but one possible gauge of the future biome distribution in the region. We compared the agreement between MC2 future output for biome and the derived "future" biome benchmark from the CA cells.

**3. Results**

*3.1. Differences in Calibration and Future Output*

Data output from the model is driven by the choice of input (historical datasets and GCM datasets), model dynamics set during calibration, and model behavior inherent to MC2 itself. Although we have no direct method to isolate the influence of CA cells on calibration and future projections, spatial and temporal differences between the two calibrated models BME-only and BME-plus can be compared.

The final calibrations for BME-only and BME-plus include notable differences. Parameter settings for the best match of each model to benchmark data differed between BME-only and BME-plus (Table 1). Both NPP and PFT distribution vary between the two calibrations. Historical averages of NPP were compared for the two models over the years 2000–2009, the same years over which the MODIS benchmark is averaged. Although NPP levels are comparable between BME-only and BME-plus, both are generally higher than the benchmark (Figure 3). NPP averaged over all biomes for the BME region is higher for BME-only, which is rooted in a larger area of forest PFTs compared to BME-plus (Figure 4). Forest growth was constrained in BME-plus by CA regions: adequate forest growth in

BME prompted too much forest growth in the CA regions (particularly in the southwest), resulting in a trade-off between forest levels within BME and outside of BME (Figure 4c). These initial configurations for NPP and forested area have a significant impact on the future output in both BME-only and BME-plus.

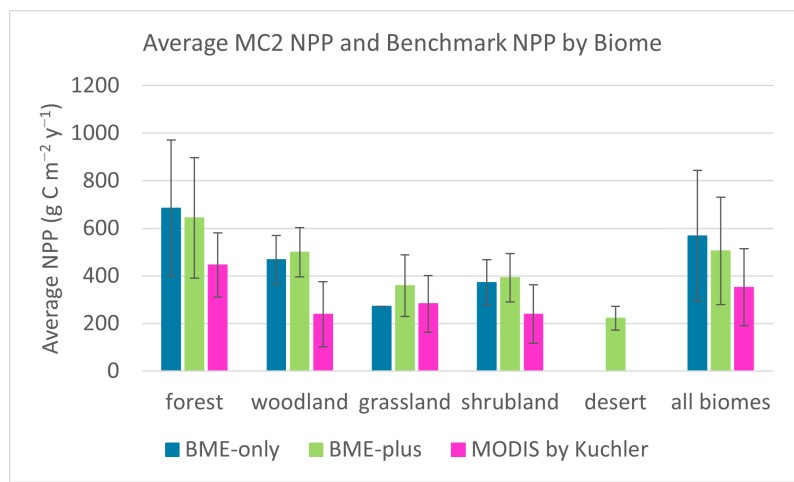

**Figure 3.** Average net primary production (NPP) for the BME-only calibration and the extended calibration domain of BME-plus by biome type, and NPP from MODIS [32] by biome type according to the Küchler dataset [33]. Averages are over the BME region and for the years 2000–2009. Error bars are standard deviations.

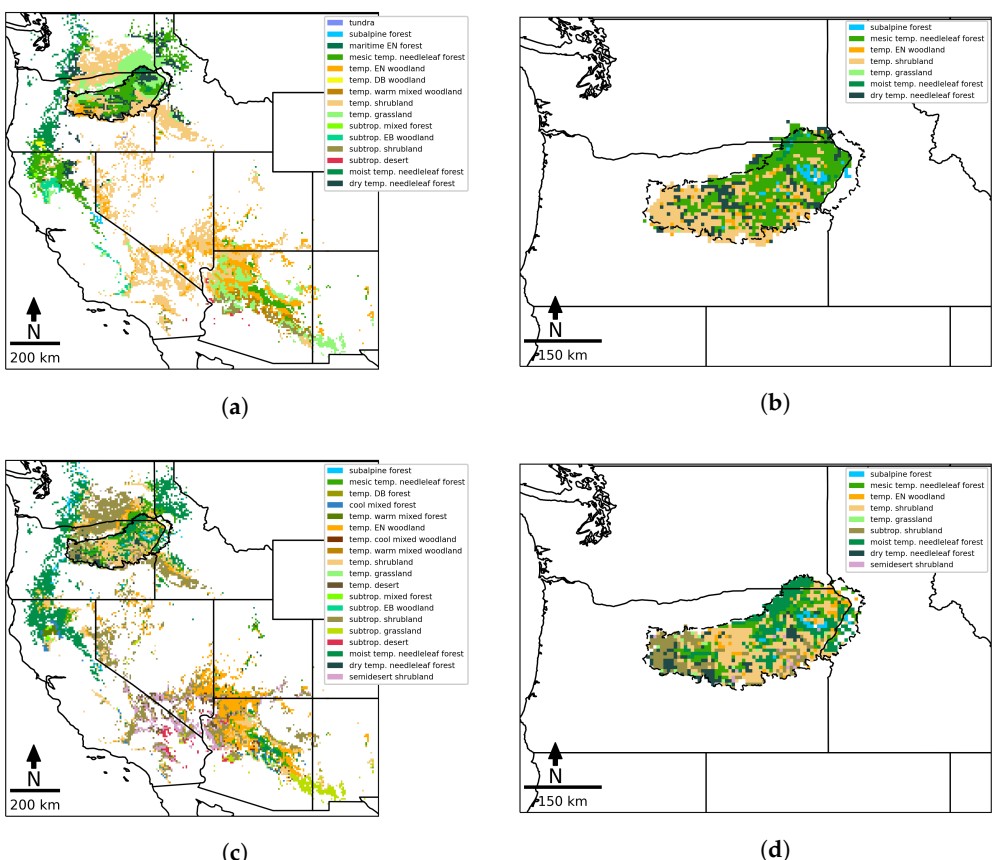

**Figure 4.** Plant functional type (PFT) for benchmark compared to study area. (**a**) PFT for Küchler vegetation benchmark [33]. (**b**) Final calibration of PFT for BME-only. (**c**) Final calibration of PFT for BME-plus. (**d**) A closer view of the BME-only region for BME-plus. In the two calibrations, PFT output is the mode of PFT for the years 1983–2012.

Model dynamics set at calibration affect the spatial and temporal output into the future. The differences in NPP resulting from calibration differences persist over time when averaged over all simulations (Figure 5). Initial differences in PFT distribution as shown in Figure 4 for calibration and in Figure 6 at full-grid resolution are also maintained over time. The amount of forest PFTs increases in both BME-only and BME-plus, although BME-only maintains a larger distribution of forest PFTs into the end of the century. As examples, the PFT modes from 2070–2099 for the simulations driven by the GCMs HadGEM2-ES and NorESM1-M are shown in Figures 7 and 8. When tracked by decade, the overall trend of increase in forest and decrease in shrubland is similar in both BME-only and BME-plus (Figure 9a–d). However, the absolute difference in these changes changes over time. Around the middle of the century, the difference between the amount of temperate shrub and dry, mesic, and moist temperate needleleaf forest in BME-only and BME-plus is at a maximum but by the end of the century, the differences are lessened (Figure 10a,b).

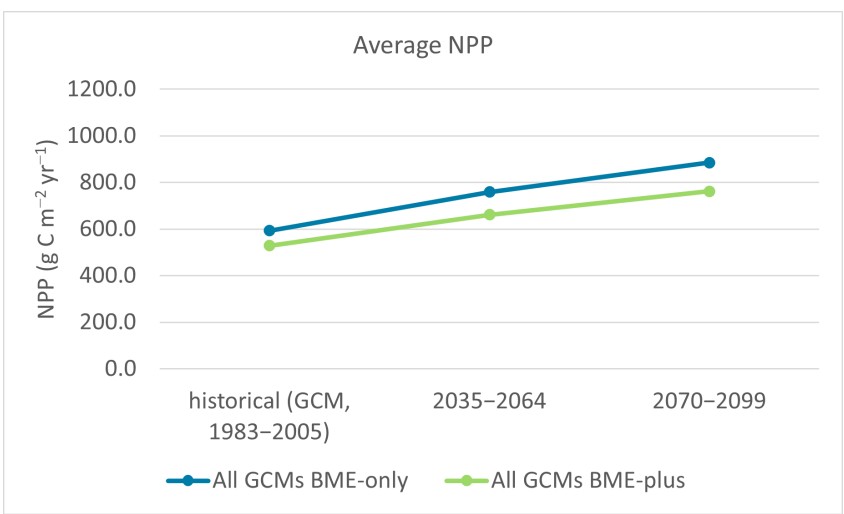

**Figure 5.** Changes in NPP over time averaged over all simulations driven by the global climate models (GCMs) for BME-only and BME-plus.

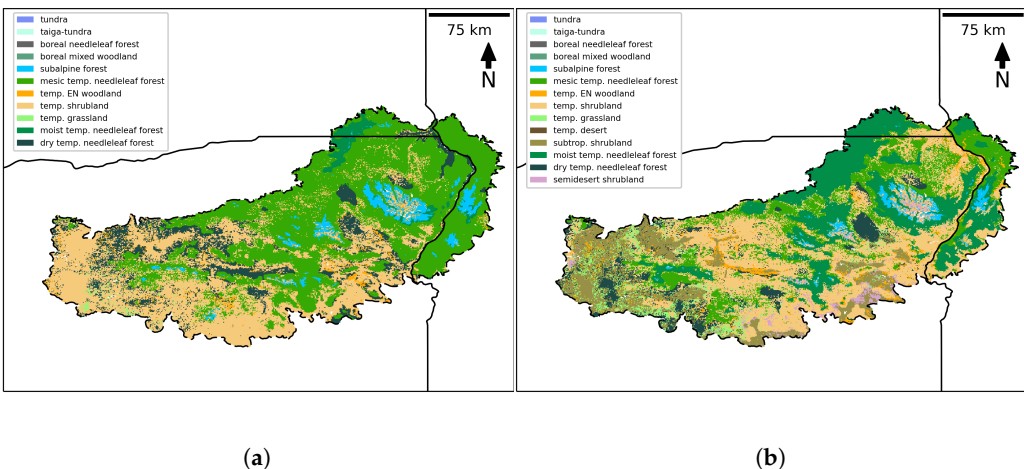

(**a**)                          (**b**)

**Figure 6.** Comparison of historical PFT mode for the two model calibrations: BME-only (study area) and BME-plus (study area plus CA cells). (**a**) PFT mode from 1983 to 2012 for the BME-only calibration. (**b**) PFT mode from 1983 to 2012 for the BME-plus calibration.

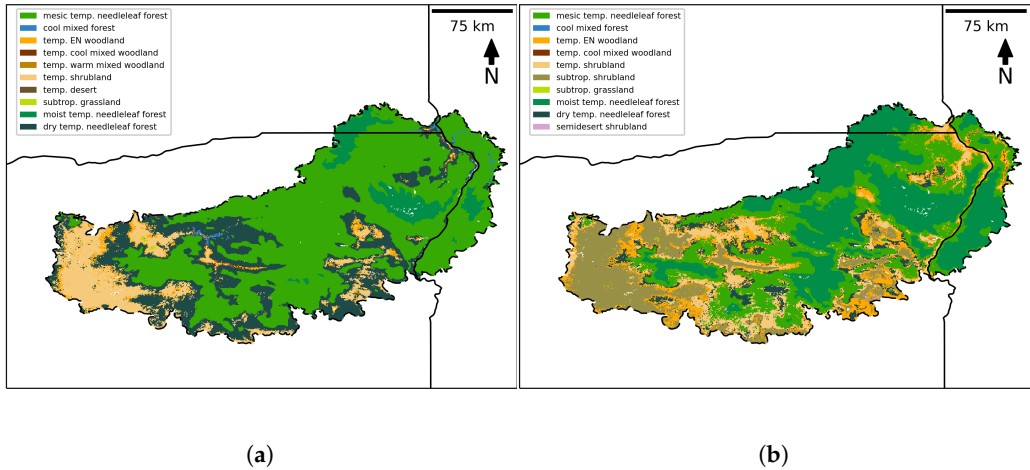

(**a**)                                                                                          (**b**)

**Figure 7.** Comparison of PFT mode for the simulation driven by the GCM HadGEM2-ES for the two model calibrations: BME-only (study area) and BME-plus (study area plus CA cells). (**a**) Mode for 2070–2099 for BME-only. (**b**) Mode for 2070–2099 for BME-plus.

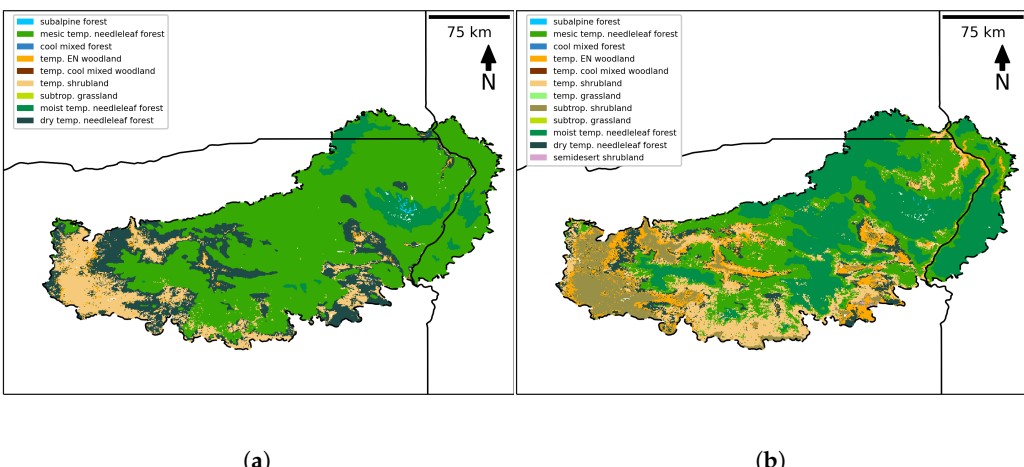

(**a**)                                                                                          (**b**)

**Figure 8.** Comparison of PFT mode for the simulation driven by the GCM NorESM1-M for the two model calibrations: BME-only (study area) and BME-plus (study area plus CA cells). (**a**) Mode for 2070–2099 for BME-only. (**b**) Mode for 2070–2099 for BME-plus.

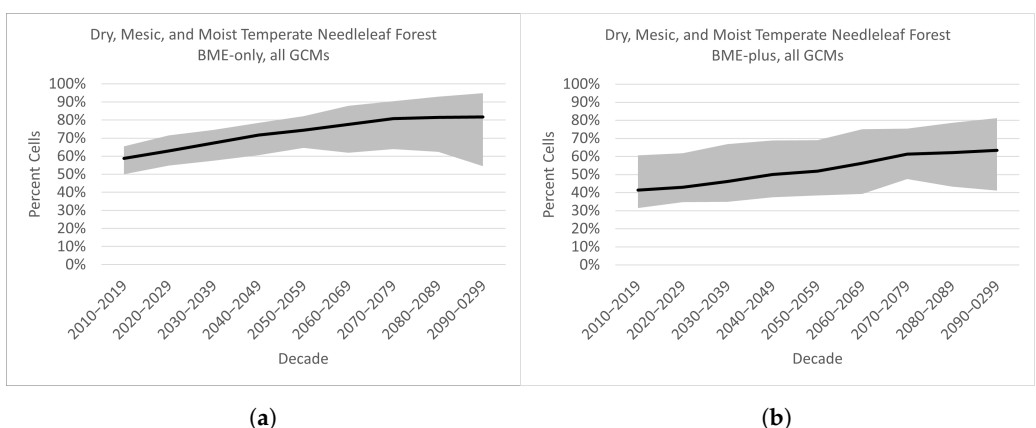

(**a**)                                                                                          (**b**)

**Figure 9.** *Cont.*

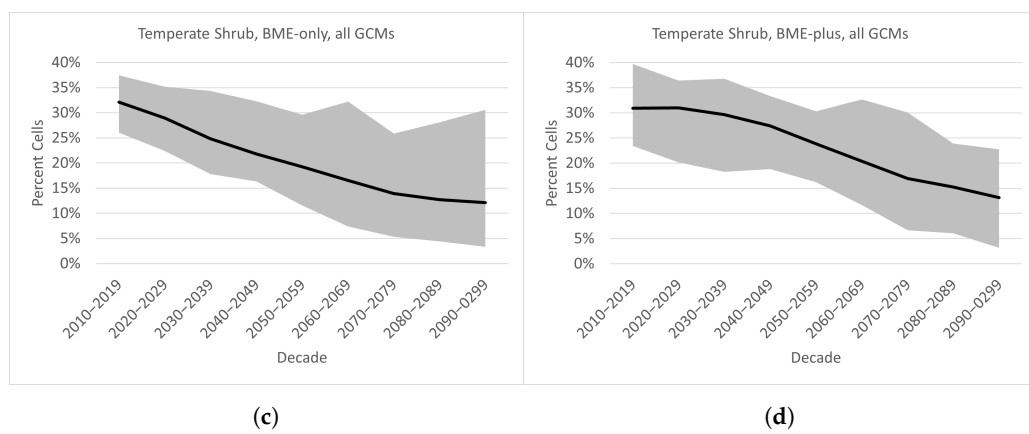

(**c**)                                                                                     (**d**)

**Figure 9.** Selected PFTs averaged by decade for BME-only and BME-plus. Percent forest (dry, mesic, temperate) in BME-only (**a**) and BME-plus (**b**). Percent temperate shrub in BME-only (**c**) and BME-plus (**d**). Black line indicates the average difference over all GCM-driven simulations; gray region represents the max and min differences over all GCM-driven simulations.

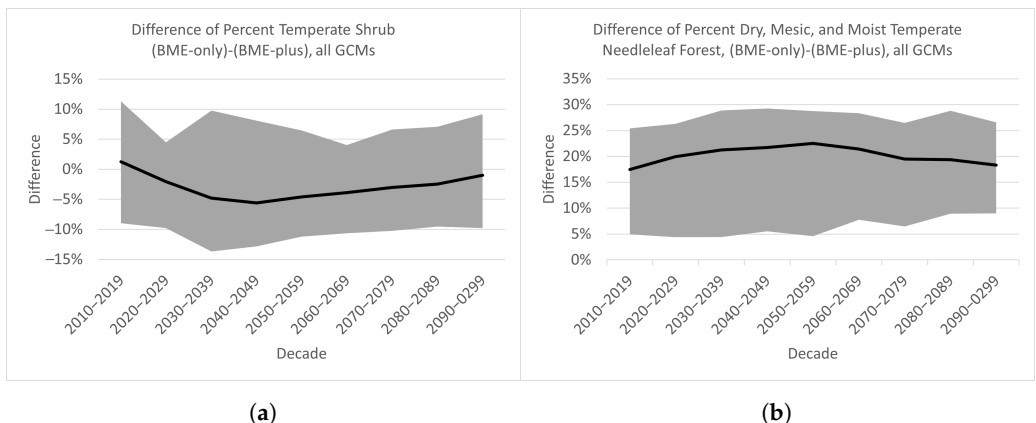

(**a**)                                                                                     (**b**)

**Figure 10.** Differences between percent of PFT in BME-only and BME-plus by decade for (**a**) temperate shrub and (**b**) dry, mesic, and moist temperate needleleaf forest. Black line indicates the average difference over all GCM-driven simulations between BME-only and BME-plus; gray region represents the max and min differences over all GCM-driven simulations.

### 3.2. Factors That Drive the Differences

As a metric of sensitivity to climate, we examined the amount of PFT change with respect to temperature change. In future projections, BME-plus PFTs are less similar to historical PFT distribution than BME-only, although both become less similar to historical PFT distribution with greater temperature changes (Figure 11). While the amount of PFT change varies across simulations, overall, the effect of the PFT change with the increased temperature change is greater for BME-plus (Figure 12a,b). Future shifts in precipitation are not correlated with elevated PFT variability (Figure 12c,d). For both temperature and precipitation, the GCM input determines the range of variation in PFT, while calibration affects the overall amount of PFT change, which, in this case, is greater for BME-plus.

As described above, we generated a "benchmark" for future projections of PFT by finding the PFT mode of CA cells according to their Küchler classification for each GCM-driven simulation and assigning the result to each BME cell as the future PFT for the given location. Figure 13 shows a count of the number of GCM-driven simulations for which the mode of CA cells is desert, shrubland, or grassland for each BME cell in (a) and a count of the number of GCM-driven simulations for which the mode of CA cells is forest for each BME cell in (b). The modeled PFT mode for each GCM-driven simulation both at mid-century (2035–2064) and the end of the century (2070–2099) was also mapped (Figure 14a–d). Unlike modeled projections for BME-only and BME-plus, the "benchmark"

projection in Figure 13 indicates decreased forest in the future and increased shrubland, grassland, and desert in the BME region. Both calibrations trend away from the analogue benchmark. Rather than progressively approaching the projected "benchmark", BME-plus departs from it to a lesser degree.

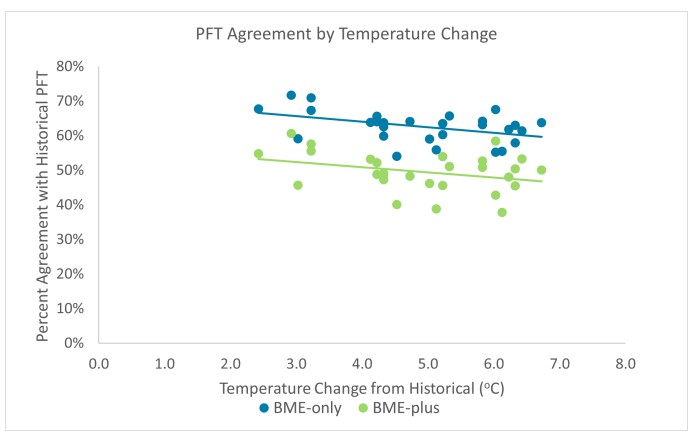

**Figure 11.** Percent PFT mode at the end of the century unchanged compared to historical mode (1983–2012) of PFT for BME-only and BME-plus per temperature change from historical temperature. Each GCM is represented as a single dot.

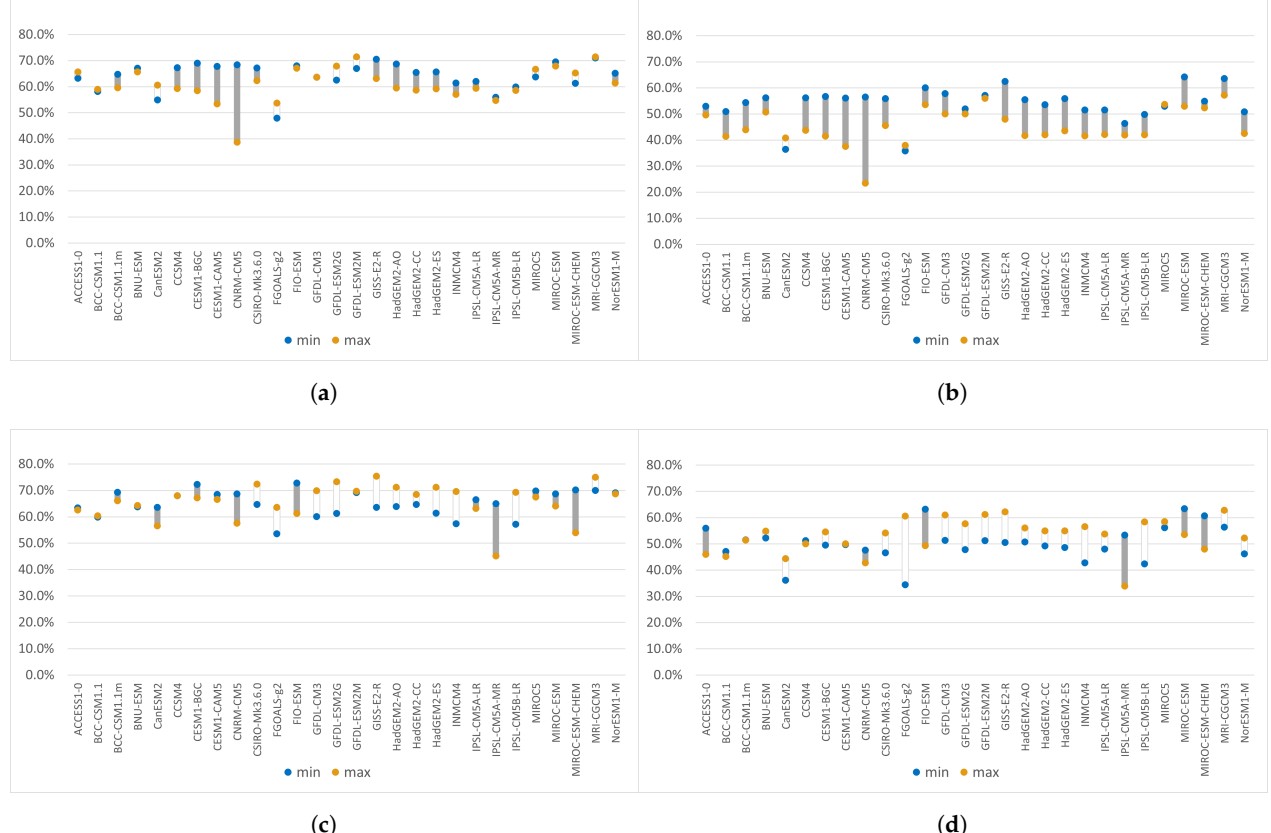

**Figure 12.** Percent PFT mode at the end of the century unchanged compared to historical (1983–2012) PFT for the 28 GCM-driven simulations by mean temperature difference from historical ((**a**), BME-only and (**b**), BME-plus) and mean precipitation difference from historical ((**c**), BME-only and (**d**), BME-plus). Blue dots are the lowest quarter (min) of temperature or precipitation change; yellow dots are the highest quarter (max) of temperature or precipitation change.

The agreement between the projected PFT for each GCM-driven simulation for BME-only and BME-plus is compared with the generated "benchmark". Agreement is always less than 50%, although the percent of cells that match with the generated "benchmark" varies between mid- and late-century (Figure 15). Greater amounts of forest in the BME-only calibration cause the BME-plus output to match the analogue benchmark (Küchler) to a slightly greater extent than BME-only. The differences shift somewhat by late century (Figure 15b), which may be due to the differing rates of forest growth in BME-only and BME-plus projections.

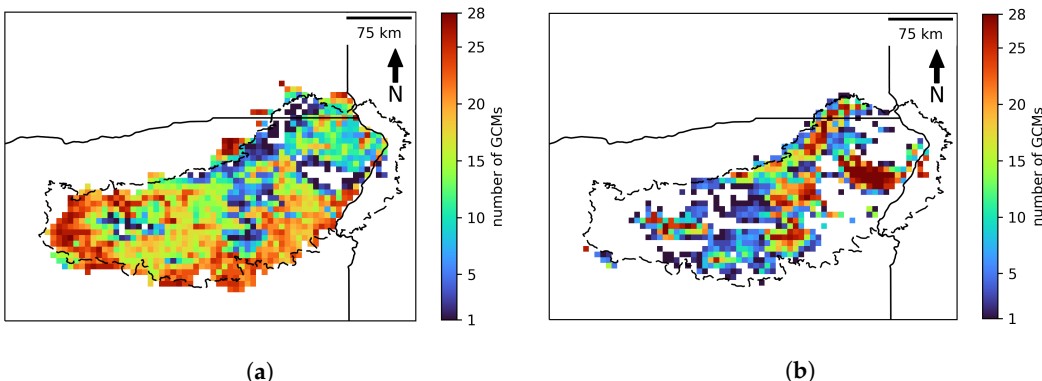

(**a**) (**b**)

**Figure 13.** Mode of the set of CA cells as overlaid by Küchler biome [33] for each BME cell. Map on the left (**a**) is a count of GCM-driven simulations, where the biome mode is either desert, shrubland, or grassland, and the maps on the right (**b**) are a count of GCM-driven simulations, where the biome mode is forest.

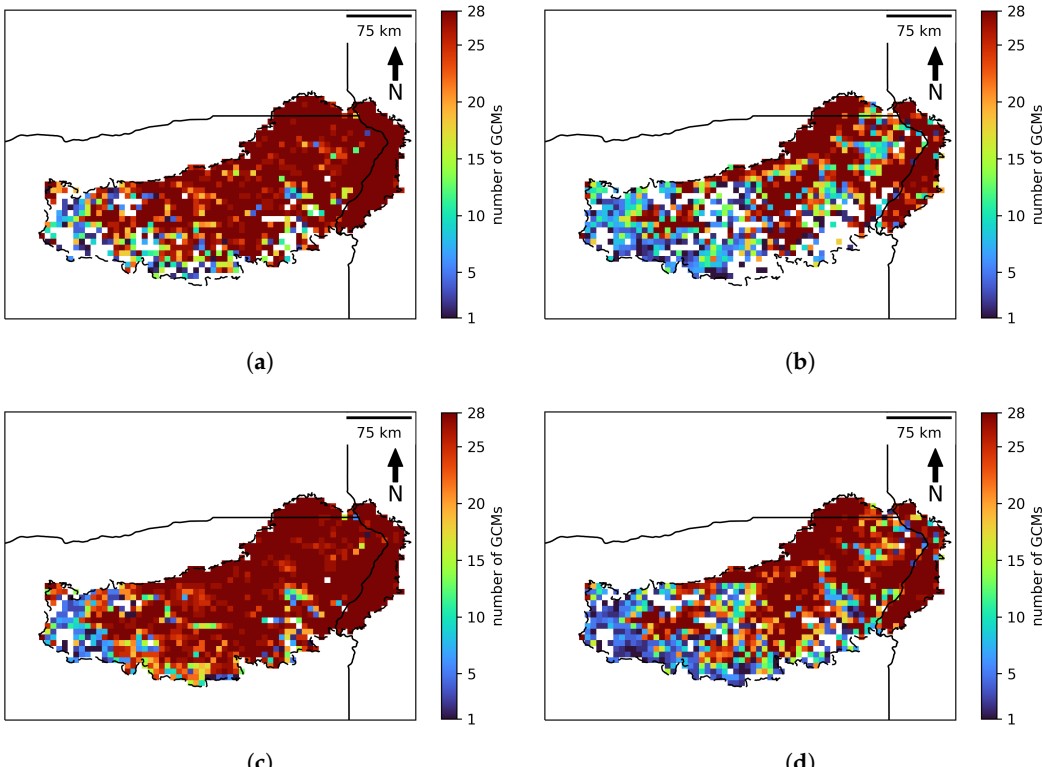

(**c**) (**d**)

**Figure 14.** Counts by pixel of the number of future simulations driven by each of the 28 GCMs, where the pixel is projected to be forest. We contrast BME-only (**a**) and BME-plus (**b**) for mid-century (2035–2064) and BME-only (**c**) and BME-plus (**d**) for end-century (2070–2099).

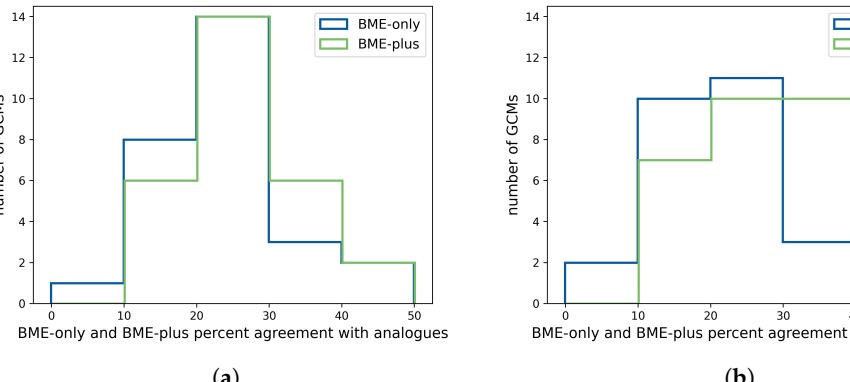

**Figure 15.** Percent of BME cells that agree with the analogue benchmark [33] per GCM, mid-century shown in (**a**) and late-century shown in (**b**).

## 4. Discussion

We showed significant differences between the future model output for BME-only and BME-plus that suggest model responsiveness to not only the GCM input but parameter choices dependent on the calibration area. For example, on average, there is ∼13% difference between the amount of vegetation type change by the end of century between BME-only and BME-plus (Figure 11). The trajectory of NPP and PFT change over time illustrates that the initial parameter choices at calibration can have a relative effect on two cognate calibrations, with the GCM input distinguishing the overall trend direction for both calibrated models. Thus, the model could be tuned to greater climatological variation by calibrating with CA cells. Ideally, unique responsiveness to climate in the study area would occur, as the modeled study region experiences climate unknown historically but represented by CA regions included in the calibration. The distribution of vegetation by the end of the century in our output is at least partly a consequence of calibration differences (Figures 6–8); a model with identical vegetation output historically that evolves to different distributions by the end of the century would more accurately capture responsiveness to climate change. In our study, we observe greater responsiveness by MC2 PFTs to climate for greater temperature differences historically, but we do not observe sensitivity to temperature and precipitation at a threshold of novel climate in the study region as would be a more realistic case. Furthermore, as demonstrated in Figure 9, extra study areas included in calibration can have an effect that itself changes over time in the evolution of the model output. Model validation efforts have shown that sensitivity to different climate forcing datasets can cause variation in output to be as large as the difference between the model and observation [7]. While we cannot definitively distinguish the influences of GCM climate variation on vegetation distribution from the influences of CA cells on calibration and future output, we demonstrated that MC2 responds to CA cell inclusion at a level discernible from mere GCM climate variation.

In a study of an individual-based forest model estimating carbon fluxes, model performance was compared for numerical calibration versus a filter-based approach [37]. The numerically calibrated model performed better overall, although the filter-based model produced better parameter values for individual processes. We find a loosely parallel result in our work: while including CA cells in calibration may have reduced the accuracy of the calibration to historical benchmarks, the overall performance of the model calibrated with CA cells was more responsive at the level of climatic change. Explained differently, better parameter values for processes historically do not necessarily lead to an overall well-performing model. These findings may reveal a coarse-graining phenomenon, where there is some loss of information in the calibration process with the inclusion of CA cells that nonetheless pushes the model toward more robust results at a different scale [9].

Including a large number of CA cells compared to the study area cells was believed to be important to capturing the range of potential future climate and its influence. We

considered that the range of climate as represented by the vegetation of the CA cells was more important than the number of CA cells. However, the actual influence of the number of CA cells would be a matter for future study. At this time, the complexity of model behavior occludes the effective isolation of the climate-analogue influence on the model parameters and output. However, the increased sensitivity of the vegetation type change to temperature in our study merits some explanation. With respect to the functionality of MC2, we consider that the adjustment of parameters in the calibration of BME-plus, which accounted for a greater range of vegetation types, set values in the internal processing of the model closer to the vegetation-type thresholds. Consequently, greater variation of vegetation-type output occurred as temperature varied. Although the biogeography module of MC2 determines the vegetation type according to limitations and thresholds, the vegetation type also influences fire behavior, which may have had additional effects on the vegetation-type change. Studies analyzing sensitivity to climate or climate-related factors, such as drought, might similarly produce variable responsiveness that changes with the inclusion of climate-analogue areas. For example, model projections informing forest vulnerability to drought and fire [6] and watershed management [4] are likely to be affected.

By creating a "benchmark" of future vegetation-type distribution from climate-analogue data, we were able to recognize not only the differences between calibration with and without CA cells, but we observed that the results of both calibrations trended away from the future "benchmark" distribution (Figures 13 and 14). This in itself suggests that the application of climate-analogue analysis to a study area can potentially provide a gauge of model behavior.

*Limitations*

We acknowledge the inherent uncertainty in the benchmark and input datasets (i.e., NPP, biome, vegetation type, and GCM input) as well as the model MC2 itself. However, as our study focused on comparative calibration methods, these uncertainties likely affect our various model outputs similarly. For this reason, we do not expect that the uncertainty of the GCM input data is a driver of difference between the BME-only and BME-plus output.

Overall, MC2 has moderate model skill in capturing present-day vegetation. For our study, soil differences between the study area and climate-analogue regions could cause vegetation-type differences that are not due to climate. MC2 models vegetation in terms of PFT, and yet, as paleoecological studies have demonstrated, species respond individually to environmental change, and the response of a plant community is the sum of these changes [38]. Therefore, MC2 is limited, in that it may not be sensitive to climate-dependent species drift. While the BME region is diverse, it serves as one example of how climate analogues in calibration affect the model output; results may, and likely would, vary by region.

As we found the influence of calibration with climate analogues to affect vegetation type and vegetation growth, the delineation of climate influence from other parameters should be better understood in modeling attempts with climate analogues. The influence of fire and other non-climatic parameters essential to the functionality of MC2 are complexities that certainly are pertinent to model output. However, we did not investigate these factors in this study, as their impact may be more relevant at a finer scale.

We recognize that the climate-analogue determination itself relies on assumptions and guesses about the future, which is a limitation in climate-analogue studies. While some have found the quality of climate analogues to vary at different scales of modeling [39], others have noted potential limitations of their usefulness. For example, the climate-analogue approach is applicable only if climate is the main driver of differences between current and analogue site pairs [40]. In general, climate-analogue rendering relies on a few climatic factors, where in reality, ecological and agricultural variables may be influential as well [41]. Our method of selecting CA cells on the basis of average precipitation and temperature values is a general approach and may not capture, for example, the seasonality

of climate analogues. We do not suggest a specific ratio or range of the climate-analogue area to include for a model calibration. The choice of the calibration area for ecological niche models and species distribution models has a significant impact on the model results and implications: not only climate but other biotic and abiotic factors can affect dispersal and colonization given various regions [1]. For hydrological models, both space and time influence the water balance simulations via the calibration period and climate conditions related to those parameters [2]. Therefore, the determination of an appropriately relevant calibration area may be specific to a given study, with climate-analogue-area inclusion at least in some form.

## 5. Conclusions

Including climate-analogue areas external to the study region in the model calibration altered future model projections in both space and time, affecting model responsiveness to climatic inputs. Calibrating BME-plus to a wider climatic range and vegetative distribution compared to BME-only led to increased PFT dynamism.

Some differences, such as higher NPP and greater forested area in BME-only, were maintained from calibration through future projections. Greater dissimilarity from the historical PFT distribution in BME-plus was a direct consequence of calibrating with CA cells. The influence of GCM input altered the ranges of PFT variability from projection to projection, and a similar pattern of variation was observed in both BME-only and BME-plus. We conclude that the GCM input is behind general patterns of PFT distribution seen in future projections as well as the models, becoming more similar toward the end of the century. However, parameters set at the calibration determined the absolute amount of PFT change in BME-only versus BME-plus and the differences between the two models that were maintained over time. We therefore acknowledge the importance of considering calibration with CA areas beyond the study region, while recognizing a need to better understand the precise influence of CA areas on future projections.

With improvement in the traceability of parameter influence throughout a model's evolution, a more direct measurement of the climate-analogue influence may be possible. We submit that further work on the inclusion of climate analogues is a relevant consideration for regional impact modeling.

**Author Contributions:** Conceptualization, E.K., J.B.K. and G.S.P.; Data curation, G.S.P.; Formal analysis, E.K., J.B.K. and R.D.; Investigation, E.K. and R.D.; Methodology, E.K. and J.B.K.; Project administration, J.B.K.; Resources, G.S.P., Software, J.B.K.; Supervision, J.B.K.; Visualization, E.K.; Writing—original draft, E.K.; Writing—review and editing, E.K. and J.B.K. All authors have read and agreed to the published version of the manuscript.

**Funding:** This research was supported in part by an appointment to the United States Forest Service (USFS) Research Participation Program administered by the Oak Ridge Institute for Science and Education (ORISE) through an interagency agreement between the U.S. Department of Energy (DOE) and the U.S. Department of Agriculture (USDA). ORISE is managed by ORAU under DOE contract number DE-SC0014664. All opinions expressed in this paper are the author's and do not necessarily reflect the policies and view of USDA, DOE, or ORAU/ORISE.

**Data Availability Statement:** The MC2 code is available online at [25]. Input and benchmark datasets are available as referenced in the text. Additional data are available on request from the corresponding author. The data are not available from a public repository due to the large size of the output and climate-analogue datasets.

**Acknowledgments:** G.S. Pitts was supported by a joint venture agreement between the USDA Forest Service Pacific Northwest Research Station and Oregon State University, under agreement 22-JV-11261954-080.

**Conflicts of Interest:** The authors declare no conflict of interest. The funders had no role in the design of the study; in the collection, analyses, or interpretation of data; in the writing of the manuscript; or in the decision to publish the results.

## Abbreviations

The following abbreviations are used in this manuscript:

| | |
|---|---|
| MDPI | Multidisciplinary Digital Publishing Institute |
| DOAJ | Directory of open access journals |
| BME | Blue Mountains Ecoregion |
| CA | Climate analogue |
| CMIP6 | Coupled Model Intercomparison Project Phase 6 |
| DGVM | Dynamic global vegetation model |
| GCM | Global climate model |
| MFRI | Mean fire return interval |
| NPP | Net primary production |
| PFT | Plant functional type |
| RCP | Representative Concentration Pathway |

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
