# Peer review of "“What’s Past Is Prologue”: Vegetation Model Calibration with and without Future Climate"

_land, doi:10.3390/land12061121_

Round 1

Reviewer 1 Report

This study demonstrated an approach where a vegetation model is subjected to two calibrations: conventionally to the study area and separately to the study area plus additional areas representing analogues of potential future climate and applied the dynamic vegetation model MC2 to a mountainous ecosystem in the Pacific Northwest USA. The results of this study may contribute to address the deficiency of model calibration with respect to climate analogues. However, there are some concerns that the authors should address before it can be considered for publication.

(1) The introduction is general and unspecific. The gaps and motivations of this study need to be further introduced and discussed.

(2) I suggest the authors add more information about data, such as data availability and access.

(3) More mechanism explanations should be added to further explain the reason for the increased sensitivity of vegetation type change to temperature.

(4) In order to further highlight the innovation of this article, it is better to compare the results of this study with other studies on model calibration.

(5) A paragraph of limitation discussion should be added to clarify the limitation or uncertainty of data and methods in this current study. For example, the uncertainty of remote sensing data including NPP and land use data (Shen et al., 2021, 2022) may affect the research results.

(6) In figures 1, 4, 6, 7, 8, 13, 14, add a north arrow and scale bar.

References:

Asymmetric impacts of diurnal warming on vegetation carbon sequestration of marshes in the Qinghai Tibet Plateau. Global Biogeochemical Cycles, 2022, 36: e2022GB007396.

Reviewer 2 Report

Dear Editor,

The manuscript I received ‘’What’s Past is Prologue: Vegetation Model Calibration With and Without Future Climate’’ discusses an approach to calibrate a vegetation model for climate change scenarios. The authors propose that traditional models are calibrated for historical conditions, but to accurately project future impacts, the model should also be calibrated for possible future climate conditions. They demonstrate this approach using a dynamic vegetation model in a mountainous ecosystem in the Pacific Northwest USA.

The study compares the conventional model calibration with the extra-study-area calibration and future projections. The results show that excluding areas representing potential future climate may be an important omission in model calibration, making the inclusion of such areas a decisive consideration in climate change impacts simulations. The study highlights the importance of considering future climate scenarios in model calibration to accurately project future impacts on ecosystems.

The article is interesting and the approach to the problem is innovative. However, since I am not a native English speaker, I will not comment on the language. Below are comments and questions about the article.

Line 31. What is the SSP5-8.5 pathway and what kind of climatic changes could occur under this pathway?

Line 26-37. What are some factors that can cause strong impacts on climate in addition to high-emissions and land use change?

Line 36-37. Why is there a lack of thorough discussion about this aspect of model calibration in scientific literature?

Line 40. How have DGVMs been applied globally, and can you provide an example?

Line 41-49. Why is improving model calibration methods important for improving the plausibility of future projections for climate change impacts?

Line 78 and others – please add the authors of the Latin species names to the manuscript when giving full species name.

Line 90-123. How accurate are the climate input data from the 28 GCMs used in the study? And are there any known limitations or weaknesses of the MC2 model, and if so, how are they addressed in the study?

Line 126. What is the spatial and temporal resolution of the MC2 model?

Line 171-182. Are there any limitations or assumptions made in the model evaluation process?

Discussion. Questions in general

What is the ideal scenario for unique responsiveness to climate in the study area?

Is there a sensitivity to temperature and precipitation at a threshold of novel climate in the study region?

How significant are the differences between future model output for BME-only and BME-plus?

What is the potential limitation of climate analogue rendering?

What could cause vegetation type differences that are not due to climate?

What complexities are pertinent to model output, but were not investigated in this study?

Reviewer 3 Report

Overall, I feel that this is a very interesting paper that addresses a potential shortcoming in how land cover models have been applied to climate simulations generated by GCMs.  They show that the information included in the vegetation models can have a strong influence on these models' predictions, and make a strong argument that information that represents future conditions, and not just local historical conditions, should be used for these model.  While this is generally a strong paper, there are a few points that could use edits or further clarifications:

Major Comments:
In the applied methods, the authors generate more than 6 times the number of CA pixels than BME pixels to be used in the MC2 for the BME-plus experiment.  Can you comment on the balance of this dataset and how including so much more data from the CA may affect the presented results?

The authors state that including all data identified in the CA pixel selection process was too computationally expensive to use, and also mention the computational performance of your code in several other places in the manuscript.  Could you describe the software and hardware used for these processes in more detail?  It would be particularly useful for others to know how the computationally expensive parts of your process was performed so that more efficient methods, or ones that parallelize better, could be proposed in the future.   

Minor Comments:

Would be helpful in the figures if the lines for the study domain were in a different style than those of the state boundaries.

The titles in figure 12 are probably not needed.
